# Estrogen Receptor Beta Agonist Influences Presynaptic NMDA Receptor Distribution in the Paraventricular Hypothalamic Nucleus Following Hypertension in a Mouse Model of Perimenopause

**DOI:** 10.3390/biology13100819

**Published:** 2024-10-12

**Authors:** Garrett Sommer, Claudia Rodríguez López, Adi Hirschkorn, Gianna Calimano, Jose Marques-Lopes, Teresa A. Milner, Michael J. Glass

**Affiliations:** 1Feil Family Brain and Mind Research Institute, Weill Cornell Medicine, 407 East 61st Street, New York, NY 10065, USAcsrodriguezlopez@gmail.com (C.R.L.); joseclopes@gmail.com (J.M.-L.); 2Center for Translational Health and Medical Biotechnology Research (TBIO)/Health Research Network (RISE-HEALTH), ESS, Polytechnic of Porto, R. Dr. António Bernardino de Almeida, 400, 4200-072 Porto, Portugal

**Keywords:** estrogens, estrogen receptor beta, menopause, neural plasticity

## Abstract

**Simple Summary:**

As women transition to menopause (i.e., perimenopause), they become more susceptible to hypertension. Animal studies using a mouse model of peri-menopause (peri-AOF) have revealed that hypertension susceptibility is associated with increased postsynaptic glutamatergic NMDA receptor plasticity in the paraventricular hypothalamic nucleus (PVN), a brain area critical for blood pressure regulation. The aim of this study was to determine if presynaptic NMDA receptors also play a role in neural plasticity in peri-AOF hypertension susceptibility. For comparison, males were also studied. Following slow pressor Angiotensin II (AngII), both peri-AOF and male mice became hypertensive; however, peri-AOF females showed higher cytoplasmic NMDA receptor levels. To determine the involvement of estrogen signaling in AngII-induced hypertension, an estrogen receptor beta (ERß) agonist was co-administered. In peri-AOF females, but not males, activation of ERß blocked hypertension and increased NMDA receptors on the membrane of axon terminals where it would be more available for binding of glutamate. These results indicate that sex-dependent recruitment of presynaptic NMDA receptors in the PVN is influenced by ERß signaling in a mouse model of perimenopause.

**Abstract:**

Women become susceptible to hypertension as they transition to menopause (i.e., perimenopause); however, the underlying mechanisms are unclear. Animal studies using an accelerated ovarian failure (AOF) model of peri-menopause (peri-AOF) demonstrate that peri-AOF hypertension is associated with increased postsynaptic NMDA receptor plasticity in the paraventricular hypothalamic nucleus (PVN), a brain area critical for blood pressure regulation. However, recent evidence indicates that presynaptic NMDA receptors also play a role in neural plasticity. Here, using immuno-electron microscopy, we examine the influence of peri-AOF hypertension on the subcellular distribution of the essential NMDA GluN1 receptor subunit in PVN axon terminals in peri-AOF and in male mice. Hypertension was produced by 14-day slow-pressor angiotensin II (AngII) infusion. The involvement of estrogen signaling was investigated by co-administering an estrogen receptor beta (ERß) agonist. Although AngII induced hypertension in both peri-AOF and male mice, peri-AOF females showed higher cytoplasmic GluN1 levels. In peri-AOF females, activation of ERß blocked hypertension and increased plasmalemmal GluN1 in axon terminals. In contrast, stimulation of ERß did not inhibit hypertension or influence presynaptic GluN1 localization in males. These results indicate that sex-dependent recruitment of presynaptic NMDA receptors in the PVN is influenced by ERß signaling in mice during early ovarian failure.

## 1. Introduction

There are established sex differences in hypertension [1]. In humans, men are at higher risk of hypertension compared to women until mid-life, but hypertension susceptibility increases as women reach menopause [2]. Male and female divergences in hypertension may involve sex differences in chromosomal complement or gonadal hormones [3]. Given that women’s susceptibility to hypertension arises at menopause onset (e.g., perimenopause), estrogen is suspected to significantly contribute to menopausal hypertension [4].

Mechanistic investigation of hormonal influences on hypertension necessitates animal models. Significantly, sex differences in hypertension are seen in small rodents [5,6,7,8]. Rodent-based investigations into hormonal contributions to blood pressure have historically focused on the ovariectomy (OVX) model [9]. However, OVX does not parallel the gradual patterns of ovarian hormone changes seen in human menopause, particularly the lack of a phase comparable to perimenopause [10,11]. Recently, this limitation has been overcome using 4-vinylcyclohexene diepoxide (VCD) in rodents to produce accelerated ovarian failure (AOF) mimicking the erratic estrogen fluctuations and extended hormone cycles seen during perimenopause, termed “peri-AOF” [10,11,12,13].

Estrogen may influence blood pressure by acting on the renin-angiotensin system [14]. Indeed, angiotensin II (AngII) is critically involved in hypertension, and when administered using the “slow-pressor” paradigm produces a neurogenic form of hypertension associated with sympathetic activation [15]. Significantly, there are notable sex differences in AngII hypertension. Compared to males, female mice show a reduced sensitivity to the pressor effects of AngII [7,8,16]. However, following the early (peri) and late (post) stages of AOF [17], female mice show increases in blood pressure in response to AngII comparable to age-matched males.

Increases in both sympathetic activity and blood pressure in AOF females critically depend on the paraventricular nucleus (PVN) of the hypothalamus (reviewed in [9]). Significantly, co-administration of agonists of estrogen receptor beta (ERß), the major estrogen receptor in the PVN [9], has been reported to suppress AngII hypertension [18].

In addition to its classical transcriptional actions, estrogen acting at ERβ has recently been shown to have rapid effects involving the modulation of NMDA receptor-mediated postsynaptic excitatory signaling [19]. Significantly, activation of NMDA receptors has been shown to play an important role in various hypertension models, including AngII-induced hypertension in male rodents [20]. In particular, immunoelectron microscopic analysis has shown that there is an increase in plasma membrane-associated levels of the essential GluN1 NMDA receptor subunit in dendritic profiles of AngII-hypertensive male mice [20]. The increase in near plasma membrane GluN1 labeling suggests an elevation in receptors capable of being mobilized to the plasma membrane in postsynaptic sites for activation by extracellular ligands. Further, although intact female mice are resistant to the hypertensive actions of AngII, peri-AOF mice infused with AngII do show both increased blood pressure and elevated plasma membrane-affiliated GluN1 in dendritic profiles of ERß-expressing PVN neurons [18]. Significantly, increases in plasma membrane-affiliated GluN1, postsynaptic NMDA currents and AngII hypertension are all reversed by administering an ERß agonist to peri-AOF mice [18].

Emerging evidence indicates that receptor plasticity is not limited to the postsynaptic compartment. Indeed, presynaptic mechanisms have been shown to influence the development of synaptic plasticity [21,22,23,24,25,26]. Presynaptic mechanisms are particularly important in the context of the NMDA receptor, given that this protein has an established presynaptic expression [27,28,29,30,31,32]. However, there is limited evidence regarding the impact of hypertension on presynaptic NMDA receptors.

To investigate the impact of AngII on presynaptic NMDA receptor localization during peri-AOF, the present study used immunoelectron microscopy, a method with spatial resolution sufficient for the detection of protein labeling in axon terminals and synapses. To investigate the influence of ERß signaling on presynaptic NMDA receptor localization during hypertension, female mice at peri-AOF were treated with the administration of an ERß agonist. In all studies matched male mice were also tested.

## 2. Materials and Methods

***Animals***: Experimental procedures were approved by the Institutional Animal Care and Use Committee of Weill Cornell Medicine and were in accordance with the 2011 Eighth Edition of the National Institute of Health Guide for the Care and Use of Laboratory Animals. Female and male bacterial artificial chromosome (BAC) ERß-enhanced green fluorescent protein (ERß-EGFP) mice on a C57BL/6 background (N = 46) were used for these studies [33]. These mice were used to characterize the estrogen sensitivity of dendritic profiles that were contacted by GluN1-labeled axon terminals. These mice were developed by the GENSAT project (www.gensat.org (accessed on 6 Octomber 2024)) at the Rockefeller University [34]. Mice were housed 3–4 per cage, kept on a 12 h–12 h light/dark cycle and were provided with food and water ad libitum.

***AOF procedure***: To induce AOF, gonadally intact female mice (52–58 postnatal days old) were injected with VCD (130 mg/kg i.p.) in the vehicle (sesame oil). Injections were given 5 days per week over three weeks [12] (Figure 1A). The early stage of AOF (i.e., peri-AOF) is characterized by irregular and extended estrous cycles and increased plasma follicle-stimulating hormone [35,36,37].

Estrous cycles were assessed by vaginal smear cytology [38] daily for 8–10 days prior to osmotic minipump insertion and at the termination of the experiment. Male mice were also handled similarly to the females throughout the experiment. The peri-AOF mice were ~3.5 months old at the time of implantation of the minipumps and had irregular and extended estrous cycles [17,18]. Peri-AOF mice had equal proportions of estrus and diestrus phases in the saline and AngII-infused groups.

***Slow-pressor AngII administration and blood pressure recording***: Osmotic mini-pumps (Alzet, Durect Corporation, Cupertino, CA, USA) containing AngII (600 ng/kg^−1^/min^−1^) in vehicle (0.1% bovine serum albumin (BSA) in saline, termed Saline [Sal]) or the vehicle alone were implanted subcutaneously under isoflurane anesthesia [8,17,18]. Systolic blood pressure (SBP) was measured in awake mice by tail-cuff plethysmography (Model MC4000; Hatteras Instruments, Cary, NC, USA). Mice were tested two days before mini-pump implantation, followed by testing at days 2, 5, 9 and 13 after pump implantation (Figure 1B). To control for possible handling effects, mice were euthanized one day after the final SBP measurements (i.e., day 14).

***ER*ß *agonist cyclic replacement regimen:*** Mice were infused with AngII and co-administered vehicle (sesame oil) or ERß agonist diarylpropionitrile (DPN; 1 mg/kg, sc) suspended in the vehicle. A cyclical administration regimen was chosen as it replicates the pulsatile pattern of natural hormone exposure [39]. Mice were “primed” with a daily dose of DPN 2 days prior to minipump implantation and then injected with the agonist every 2–3 days for a total of 10 injections. A schematic timeline of the ERβ agonist injection regimen, coinciding with AngII induction and SBP readings is provided in Figure 1C.

***Experimental Groups:*** The cohorts of animals used in this study were the same as those in prior studies [8,17,18]. Eight groups of ERß-EGFP mice were used. Groups 1 and 2: peri-AOF female mice infused with Sal or AngII [17]; groups 3 and 4: male mice infused with Sal or AngII [8]; groups 5 and 6: peri-AOF female mice infused with AngII and injected with either Veh or DPN [18]; and groups 7 and 8: male mice infused with AngII and injected with either Veh or DPN [18]. Experiment 1 comprised Groups 1–4; experiment 2 comprised groups 5–8.

***Antibodies***: To label the EGFP reporter, a chicken polyclonal anti-GFP antibody generated against recombinant GFP [40] was used (GFP-1020; RRID: AB_10000240; Aves Lab Inc., San Diego, CA, USA). This antibody recognizes one major band at ~27 kD and immunolabels cells in brain sections from transgenic mice expressing EGFP (manufacturer’s data sheet, www.antibodiesinc.com/collections/aves-labs, accessed on 6 Octomber 2024) on Western blots. In mice not expressing EGFP, immunolabeling for the GFP antibody is absent [41,42]. Further, the fidelity of GFP labeling has been shown by corresponding immunolabeling of ERß protein as well as mRNA in mouse brains [33,43].

To label GluN1, a mouse monoclonal antibody (clone 54.1; RRID: AB_86917; BD Biosciences, San Diego, CA, USA) was used. Specificity of the GluN1 antibody has been shown by immunoprecipitation, immunohistochemistry [44,45] and in conditional knockout mice [46]. Additionally, specificity has been demonstrated by Western blot analysis of rat synaptic membranes, monkey hippocampal homogenates and HEK 293 cells transfected with cDNA encoding GluN1 [45].

***Tissue preparation***: Mouse brains were prepared and processed for pre-embedding dual immunolabeling for electron microscopy as described [42]. Following deep sodium pentobarbital (150 mg/kg, i.p.) anesthesia, mouse brains were fixed via aortic arch perfusion with ~5 mL of 2% heparin in normal saline followed by 30 mL of 2% paraformaldehyde (PFA) and 3.75% acrolein in 0.1 M phosphate buffer (PB; pH 7.4). Brains were removed from the skull and post-fixed at room temperature in 2% PFA and 2% acrolein in PB for 30 min. Whole brains were cut into 5 mm coronal blocks using a brain mold (Activational Systems, Inc., Warren, MI, USA) and then sectioned (40 µm) using a VT1000X Vibratome (Leica Microsystems, Buffalo Grove, IL, USA). Brain sections through the PVN were collected and placed in a cryoprotectant solution (30% sucrose, 30% ethylene glycol in PB) and then stored at −20 °C until immunocytochemical processing. To ensure uniformity of immunohistochemical labeling, sections from each experimental group were processed under identical immunolabeling conditions. For this, each tissue section was marked with an identifying punch code in the cortex, and sections were then pooled into single containers for processing. Samples consisted of two sections per animal (3 animals per group) from the intermediate-caudal PVN (0.70–0.94 mm caudal to bregma [47]).

***Electron microscopic dual labeling immunohistochemistry***: Reactive aldehydes were neutralized after incubating brain sections in 1% sodium borohydride in PB for 30 min followed by 8–10 rinses in PB. Next, brain sections were blocked by incubation in 0.5% bovine serum albumin (BSA) in 0.1 M Tris-buffered saline (TS) for 30 min. After TS washes, tissue then was incubated in a cocktail of chicken anti-GFP (1:3000) and mouse anti-GluN1 antibody (1:50) in 0.1% BSA/TS) for 1 day at room temperature and then 2 days at 4 °C. All incubations were performed on a shaker at 145 RPM.

For GFP immunoperoxidase labeling, sections were first incubated in goat anti-chicken IgG (1:400; Jackson ImmunoResearch Inc., West Grove, PA, USA) for 30 min and then washed in TS. Next, sections were incubated for 30 min in avidin–biotin complex at half the manufacturer’s recommended dilution (Vector Laboratories, Burlingame, CA, USA). After washing in TS, immunoreactivity was visualized by reaction for 3 min in 3,3′-diaminobenzidine (DAB; Sigma-Aldrich Chemical Co., Milwaukee, MI, USA) and hydrogen peroxide.

For silver-intensified immunogold (SIG) labeling of GluN1, GFP-labeled brain sections were incubated for 1 day in goat anti-mouse IgG conjugated with 1 nm colloidal gold particles (1:50; Electron Microscopy Sciences (EMS), Fort Washington, PA, USA) in a solution consisting of 0.01% gelatin and 0.08% BSA in 0.01 M phosphate-buffered saline (PBS). Following incubation, brain sections were washed in PBS, and then incubated in 2% glutaraldehyde in PBS for 10 min. Next, brain sections were rinsed in PBS, and then placed in 0.2 M sodium citrate buffer (pH 7.4). The bound gold particles were intensified by incubating brain sections for 7 min in a silver enhancement solution (either IntenSEM kit RPN491; GE Healthcare, Waukeska, WI, discontinued [8,17] or SEKL15 Silver enhancement kit, product #15718 Ted Pella [18]).

***Preparation for electron microscopy***: After dual immunolabeling, brain sections were post-fixed in a solution of 2% osmium tetroxide in PB for 1 hr. Next, sections were washed in PB, dehydrated through a series of ethanol incubations, followed by propylene oxide and incubated in 1:1 propylene oxide/ EMbed-812 (both from EMS) overnight. Following incubation in EMbed-812 for 2 h sections were placed between two sheets of Aclar plastic and backed at 60 °C for 3 days. Ultrathin sections (~70 nm thick) from the PVN were cut using a Leica EM UCT6 ultratome and a Diatome diamond knife (EMS). Ultrathin sections were collected on 400-mesh thin-bar copper grids (T400-Cu, EMS) and counterstained with either Uranyl acetate (EMS #22400) and Reynold’s lead citrate (EMS #17900-25) [8,17] or Uranyless^TM^ (catalog #22409, EMS) and lead citrate (catalog #22410 EMS).

***Ultrastructural data analyses***: Persons blinded to experimental conditions conducted all data collection and analyses. Images used in this analysis were the same ones acquired in our analysis of dual-labeled GluN1 and GFP-labeled dendrites in prior studies [8,17,18]. For this, 50 dual-labeled dendritic profiles were randomly selected from each PVN block and then photographed at a magnification of 13,500× using a digital camera system (version 3.2, Advanced Microscopy Techniques, Woburn, MA, USA) on a Tecnai transmission electron microscope (Tecnai 12 BioTwin, FEI, Hillsboro, OR, USA). To control for differences in antibody penetration, analyzed tissue fields were taken at the tissue-plastic interface [42]. The identification of terminals and dendrites was determined according to standard morphological criteria [48]. Terminals contained numerous small synaptic vesicles, mitochondria and sometimes formed synapses on soma and dendrites. Dendrites were identified by the presence of a post-synaptic specialization, regular microtubular arrays and mitochondria. Immunoperoxidase labeling was characterized by a diffuse precipitous electron-dense reaction product. The presence of SIG labeling was marked by punctate black electron-dense particles. Of note, none of the GluN1-labeled terminals colocalized GFP.

In the present study, all terminals with GluN1 labeling were assessed in fields that contained dual-labeled dendrites. The location of GluN1-SIG particles within terminals was determined as this has functional significance. Plasma membrane labeling corresponds to sites of receptor binding, whereas near-plasma membrane labeling likely reflects receptors available for insertion or removal from the plasma membrane [49,50]. Further, cytoplasmic labeling may identify receptors stored or in transit to other cellular compartments, as well as in the process of being degraded or recycled [51,52]. Therefore, using previously described quantitative methods [18], SIG GluN1 labeling was categorized into three subcellular compartments: (1) particles on the plasma membrane (onPM), (2) particles located within 70 nm of the PM (nearPM) and (3) particles localized to the cytoplasm (cyto). Moreover, if terminals contacted dendrites, we noted whether the dendrite contained GFP peroxidase labeling and/or GluN1 SIG particles and whether the terminals formed asymmetric synapses (excitatory type), symmetric synapses (inhibitory type) or appositions (plasma membranes juxtaposed) on the dendrites.

***Figure preparation:*** Images were adjusted for brightness, levels and sharpness using an unsharp mask in Adobe Photoshop 2020 (RRID:SCR_014199). Images were imported into Microsoft PowerPoint for Mac (version 16.77), for additional changes to brightness, contrast, and sharpness. These adjustments did not alter the original content of the raw image and were made to achieve a uniform appearance between micrographs. Graphs were made using Prism 10 software (Graphpad Prism, RRID:SCR_002798).

***Statistical analysis:*** Data were expressed as means ± SEM. Significance was set to an alpha < 0.05. A trend was determined as alpha < 0.07. Statistical analyses were conducted on Prism 10. Two-way analysis of variance (ANOVA) was followed by a post hoc Tukey’s HSD (SBP data) or a Fisher’s LSD (EM dual-labeling data).

## 3. Results

### 3.1. Blood Pressure in AngII-Infused Peri-AOF and Male Mice without or with Cyclic ERß Agonist Administration

Prior to implantation of the osmotic minipumps, there were no significant differences in SBP between any of the peri-AOF mice and male mice (i.e., baseline measurements).

As previously reported [8,17], SBP was recorded in peri-AOF female mice and in male mice infused with either Sal or AngII (Figure 1B). Two-way ANOVA showed a significant main effect of Sal/AngII administration (F(3, 21) = 7.135, *p* = 0.0017) and a Sal/AngII administration x time point interaction (F(3, 21) = 8.488, *p* = 0.0007). Post hoc analysis revealed an increase in SBP on day 13 post-implantation compared to day 0 in both AngII-infused peri-AOF female (*p* = 0.016) and male (*p* = 0.003) mice (Figure 2A). In contrast, blood pressure was not elevated in Sal-infused peri-AOF female mice on day 13 compared to day 0. There were no significant differences in SBP in male mice infused with saline on day 0 vs. day 13 (Figure 2A).

As previously reported [18], SBP was recorded in peri-AOF female mice and in male mice infused with AngII and co-administered with Veh or DPN on a cyclic regimen (Figure 1C). Two-way ANOVA showed significant main effects of Veh/DPN administration (F(3, 40) = 4.376, *p* = 0.009) and of time point (F(1, 40) = 111.2, *p* < 0.0001), as well as a significant Veh/DPN administration x time point interaction (F(3, 40) = 9.57, *p* < 0.0001). In agreement with the above results, post hoc tests showed that infusion of AngII increased SBP in both peri-AOF Veh-injected females (*p* < 0.0001) and Veh-injected males (*p* < 0.0001) (Figure 2B). However, post hoc comparisons revealed that the SBP in peri-AOF females co-administered DPN during the AngII infusion was not significantly different on day 0 compared to day 13 (Figure 2B). In contrast, SBP was significantly elevated (*p* < 0.0001) on day 13 compared to day 0 in male mice co-administered DPN during the AngII infusion (Figure 2B).

### 3.2. Presynaptic GluN1 in Mice Infused with Sal or Ang

Immunoelectron microscopy was used to investigate the relationship between hypertension and presynaptic NMDA receptor subcellular localization. Given that GluN1 is essential in the formation of most NMDA receptor variants [53], we immunolabeled GluN1 as a stand-in for NMDA receptors.

In all groups, GluN1-labeled axon terminals were typically ovoid, usually ranged between 0.4 and 0.6 µm in diameter and contained numerous small clear synaptic vesicles and mitochondria (Figure 3A–D). Collapsed across treatments (i.e., Sal and AngII), a total of 796 GluN1-labeled axon terminals were sampled from the PVN of peri-AOF female (n = 477) and male (n = 319) mice. Axon terminals were further subdivided by females infused with either saline (n = 213) or AngII (n = 264) as well as males infused with either saline (n = 165) or AngII (n = 154).

***Subcellular distribution of GluN1***: Because the EM experiments for peri-AOF female and male mice were conducted separately, direct comparisons of GluN1 SIG particle number in terminals between sexes could not be made. However, the average number of total GluN1 SIG particles per axon terminal was similar in Veh and AngII-infused peri-AOF female and male mice (Figure 3E,F). In peri-AOF females, the number of GluN1 SIGs in the on or near PM compartment of terminals was not significantly different in the Sal and AngII-infused mice (Figure 3E). However, GluN1 in the cytoplasm of terminals from peri-AOF females tended (t_6_ = 1.858; *p* = 0.064) to be lower in AngII-infused mice compared to Sal-infused mice (Figure 3E). In males, the number of onPM GluN1 SIG particles in terminals was significantly lower (*p* = 0.037) following AngII (Figure 3F).

To directly compare peri-AOF females with males, we determined the ratio of GluN1 SIG particles in each subcellular compartment out of the total number of particles in all terminals. Two-way ANOVA of the onPM compartment showed an interaction effect of sex x treatment (F(1, 792) = 4.264, *p* = 0.039) in the Sal and Ang-infused mice. Two-way ANOVA showed a main effect of sex for the near PM compartment (F(1, 792) = 23.39, *p* < 0.0001) and the cytoplasmic compartment (F(1, 792) = 24.45, *p* < 0.0001). Post hoc comparisons demonstrated that following AngII-infusion, male mice tended to have a decreased proportion of onPM GluN1 SIG particles compared to their Sal-treated male counterparts (t_6_ = 1.893; *p* = 0.059) (Figure 3G). Additionally, male mice had significantly higher proportions of nearPM GluN1 SIG particles than their respective Sal (*p* = 0.0296) and AngII-infused (*p* < 0.0001) female counterparts (Figure 3G).

In contrast, peri-AOF female mice had a greater proportion of cytoplasmic GluN1-SIGs in axon terminals compared to their respective Sal (*p* = 0.0032) and AngII-infused (*p* < 0.0001) male counterparts (Figure 3G).

Next, the partitioning ratio of GluN1 was assessed in the subset (68.37%) of terminals that formed synapses or appositions with dendritic profiles. Two-way ANOVA of the onPM compartment showed an interaction effect of sex x treatment (F(1, 530) = 4.191, *p* = 0.0411) in the Sal- and Ang-infused mice. Two-way ANOVA also showed a significant main effect of sex with respect to the nearPM (F(1, 530) = 7.974, *p* = 0.005) and cytoplasmic (F(1, 530) = 8.922, *p* = 0.003) compartments. Post hoc comparisons demonstrated that Sal peri-AOF females tended to have a higher proportion of cytoplasmic GluN1 SIG particles (t_6_ = 1.875; *p* = 0.06) than their male counterparts (Figure 3H). Moreover, AngII-infused peri-AOF females showed a significantly lower proportion of nearPM GluN1 SIG (*p* = 0.004) and a higher proportion of cytoplasmic GluN1 SIG particles (*p* = 0.019) compared to their AngII-infused male counterparts (Figure 3H).

***Synapse types formed by GluN1 terminals:*** Of the GluN1-labeled terminals that formed contacts (approximately two-thirds), the proportion forming asymmetric and symmetric synapses and appositions was determined for each group. Two-way ANOVA showed a main effect of the form of terminal contact between the four groups (F(2, 24) = 5.066, *p* = 0.015). Post hoc analyses showed that the proportions of appositions and asymmetric and symmetric synapses formed by GluN1 terminals were not significantly different between peri-AOF and male mice, regardless of AngII-infusion (Figure 3I). However, of the contacts formed by GluN1-labeled terminals, about 25–30% were asymmetric and about 35–40% were symmetric (Figure 3I).

***Dendritic targets of GluN1 terminals*:** When collapsed across all groups, nearly all of the dendrites contacted by GluN1-labeled presynaptic terminals contained GluN1 (84.35%). However, not all post-synaptic dendrites contained ERß (65.41% GluN1 only, 34.59% GluN1 + ERß). Two-way ANOVA revealed a main effect of the proportion of single- and double-labeled dendrites (F(1, 16) = 14.10, *p* = 0.002). Post hoc tests showed that the proportions of unlabeled, single and double-labeled dendritic targets were not significantly different between peri-AOF and male mice, regardless of AngII-infusion (Figure 3J).

### 3.3. Presynaptic GluN1 in Ang-Infused Mice Co-Administered DPN

Examples of GluN1-labeled axon terminals from AngII-infused peri-AOF females and males co-administered either Veh or DPN are shown in Figure 4A–D. Collapsed across treatments, there was a total of 574 GluN1-labeled axon terminals sampled from the PVN of female (n = 274) and male (n = 300) mice. From this population, axon terminals were further subdivided into AngII-infused peri-AOF females injected with Veh (n = 133) or DPN (n = 141) as well as AngII-infused males injected with either Veh (n = 130) or DPN (n = 170).

***Subcellular distribution of GluN1***: Since the EM experiments for AngII-infused peri-AOF female and male mice administered Veh or DPN were conducted simultaneously, direct comparisons of GluN1 SIG particle number in terminals between sexes could be made. Two-way ANOVA showed a significant DPN administration x sex interaction effect in the total number of GluN1 SIG particles per terminal (F(1, 570) = 8.773, *p* = 0.003) and the number of cytoplasmic GluN1 SIG particles per terminal (F(1, 570) = 4.011, *p* = 0.046). Post hoc analysis revealed that following DPN administration, AngII-infused peri-AOF female mice had a decreased number of cytoplasmic GluN1 SIG particles (*p* = 0.049) compared to their Veh-injected counterparts (Figure 4E). Conversely, AngII-infused male mice injected with Veh showed a greater total number of GluN1 SIG particles (*p* = 0.013) compared to AngII-infused males injected with DPN (Figure 4F).

Sex differences in the subcellular distribution of GluN1 SIG particles in terminals were also noted. Terminals from AngII-infused Veh peri-AOF female mice had more total GluN1 SIG particles (*p* = 0.042) and cytoplasmic GluN1 SIG particles (*p* = 0.012) compared to AngII-infused Veh males (Figure 4E,F). Terminals from AngII-infused peri-AOF mice injected with DPN had a lower number of nearPM GluN1 SIG particles (*p* = 0.028) and total number of GluN1 SIG particles (*p* = 0.032) compared to AngII-infused males injected with DPN (Figure 4E,F).

The partitioning ratio of GluN1 in each subcellular compartment for all terminals was determined for each group. Two-way ANOVA showed a main effect of DPN administration (F(1, 570) = 4.254, *p* = 0.0396) for GluN1 SIG in the onPM compartment (Figure 4G). However, further post hoc multiple comparisons did not show any significant differences between individual treatments.

Next, the partitioning ratio of GluN1 in the subset (73%) of terminals that formed synapses or appositions with dendritic profiles for all groups was determined. Two-way ANOVA revealed a main effect of DPN administration on the ratio of the GluN1 SIG onPM compartment (F(1, 417) = 9.76, *p* = 0.002). Post hoc multiple comparisons showed that AngII-infused peri-AOF female mice injected with DPN exhibited an increased proportion of onPM GluN1 SIG particles (*p* = 0.008) compared to their Veh-injected counterparts (Figure 4H). Moreover, DPN-injected peri-AOF females compared to oil-injected peri-AOF females tended to have a lower proportion of cytoplasmic GluN1 SIG particles (t_6_ = 1.886; *p* = 0.06) in axon terminals (Figure 4H).

***Synapse types formed by GluN1 terminals:*** Of the GluN1-labeled terminals that formed contacts (approximately three-fourths), the proportion forming asymmetric and symmetric synapses and appositions was determined for each group. Collapsed across all groups, the majority of terminals formed symmetric connections (57.24%), while a smaller percentage (9.03%) formed asymmetric synapses. Two-way ANOVA revealed a main effect of postsynaptic contact (F(2, 24) = 209.0, *p* < 0.0001) and an interaction effect of postsynaptic contact and treatment (F(6, 24) = 5.059, *p* = 0.002). Post hoc tests showed that GluN1 terminals from AngII-infused peri-AOF female mice injected with DPN had a significantly greater proportion of asymmetric synapses (*p* = 0.005) and a lesser proportion of symmetric synapses (*p* = 0.015) compared to their DPN-injected male counterparts (Figure 4I).

***Dendritic targets of GluN1-labeled terminals:*** In AngII-infused mice administered Veh or DPN, all but one of the dendrites targeted by GluN1-labeled presynaptic terminals contained GluN1 (99.64%). Collapsed across all groups, 40.29% of the post-synaptic dendrites that contained GluN1 also contained ERß. The proportion of single and double-labeled dendritic targets of GluN1-labeled presynaptic terminals varied by sex and treatment. Two-way ANOVA showed a main effect of the form of postsynaptic labeling (F(2, 24) = 81.82, *p* < 0.0001) and an interaction effect of the form of postsynaptic labeling and treatment (F(6, 24) = 2.659, *p* = 0.04). Post hoc analysis revealed that vehicle-injected AngII-infused peri-AOF female mice had a lower proportion of GluN1 single-labeled dendritic targets (*p* = 0.031) and a greater proportion of GluN1 + ERß double-labeled dendritic targets (*p* = 0.022) than DPN-injected peri-AOF females (Figure 4J). Veh-injected females also exhibited a lower proportion of single-labeled dendritic targets (*p* = 0.026) and a higher proportion of double-labeled dendritic targets (*p* = 0.026) than their male counterparts (Figure 4J).

## 4. Discussion

We investigated the influence of hypertension on presynaptic NMDA receptor localization during an early stage of ovarian failure. For this, dual-labeling EM was used to assess the impact of AngII infusion on labeling of the essential GluN1 subunit in axon terminals in the PVN of peri-AOF mice as well as matched males. We report an important influence of sex on the subcellular partitioning of presynaptic GluN1, with no effect of AngII on GluN1 localization (Figure 5). It was also shown that co-administration of the ERß agonist DPN selectively attenuated AngII hypertension in peri-AOF mice, which was accompanied by altered GluN1 distribution in terminals in peri-AOF mice alone. These findings indicate that sex and ERß signaling contribute to presynaptic GluN1 localization in the PVN.

Our prior studies have reported that AngII produces sexually dimorphic effects on both blood pressure and postsynaptic NMDA receptor signaling in the PVN [18,20]. Specifically, it has been shown that in AngII hypertensive male mice GluN1 is increased on the plasma membrane in dendrites of PVN neurons [20]. In contrast, intact young female mice have a blunted pressor response to AngII and do not show increased plasma membrane GluN1 [8]. However, peri-AOF mice display elevated blood pressure following AngII along with increased plasma membrane-affiliated GluN1 in ERß-expressing dendrites [17]. Further, the increased GluN1 signaling at peri-AOF is attenuated by administering DPN [18], indicating a critical role for ERß in heightened NMDA signaling in the PVN during AngII infusion at early AOF. These results indicate that AngII hypertension in both male and peri-AOF female mice is associated with increased postsynaptic NMDA receptor signaling, but only in females do hypertension and NMDA receptor plasticity involve ERß.

Beyond the postsynaptic compartment, it is increasingly recognized that NMDA receptors have actions extending to presynaptic sites as well [54,55]. Indeed, biochemical, immunohistochemical light microscopic, and electron microscopic ultrastructural studies all point to the presence of presynaptic NMDA receptors in various brain regions [31]. Additionally, functional evidence also demonstrates important roles for presynaptic NMDA receptors in synaptic signaling and plasticity mediated by various mechanisms [56].

Given this background, it is important to also consider the involvement of presynaptic mechanisms in hypothalamic NMDA receptor plasticity. Further, presynaptic mechanisms may be particularly relevant to neurogenic hypertension given evidence of pronounced NMDA receptor immunolabeling in axon terminals in the PVN [57]. However, there is little evidence regarding the involvement of presynaptic NMDA receptors in the PVN during hypertension in peri-AOF or male mice.

High-resolution immunohistochemical electron microscopy was used to assess the effect of AngII infusion on the subcellular distribution of GluN1 in axon terminals of PVN neurons of peri-AOF mice and male mice infused with AngII. It was found that AngII hypertension did not significantly affect GluN1 localization in axon terminals in either peri-AOF or male mice. However, there was an important contribution of sex to presynaptic GluN1. In males, the partitioning ratio of GluN1 in near plasma membrane locations was higher compared to peri-AOF mice. In contrast, the partitioning ratio of cytoplasmic GluN1 labeling was elevated in axon terminals in the PVN of peri-AOF compared to male mice. Neither Sal- nor AngII-infused mice showed changes in total presynaptic GluN1 labeling. Additionally, there were no differences in the number of axon terminals forming asymmetric or symmetric synapses across groups. Given that plasma membrane receptors are positioned for functional activation by extracellular ligands, these results suggest a heightened propensity for presynaptic NMDA receptor signaling in males compared to peri-AOF mice.

Estrogen acting at ERß, the predominant receptor in the PVN, has been shown to contribute to blood pressure regulation [9,58]. Estrogen-induced activation of ERβ is well known to elicit transcriptional activation; however, emerging evidence also indicates ERß rapidly modulates NMDA receptor-mediated excitatory signaling [19]. In the PVN, postsynaptic ERß appears to protect cycling females against hypertension and heightened NMDA receptor signaling associated with AngII [59]. Further, reduced estrogen signaling also contributes to the sensitivity of AOF mice to hypertension and elevated NMDA receptor signaling [18]. This contrasts with male mice who are sensitive to AngII hypertension and do not respond to ERß agonists [18]. Presumably, compared to females, male mice lack protection from AngII due to lower estrogen levels.

In view of the importance of ERß in NMDA receptor signaling during hypertension [18], we further investigated the effect of DPN administration on the localization of GluN1 in axon terminals of PVN neurons in AngII-infused peri-AOF and male mice. In peri-AOF mice, it was found that the antihypertensive action of DPN was associated with an increase in the partitioning ratio of GluN1 on the plasma membrane in axon terminals forming contacts with dendrites. In contrast, DPN did not inhibit hypertension or influence subcellular GluN1 localization in male mice. Overall, these results indicate that hypertension and levels of potentially functional presynaptic NMDA receptors in the PVN are influenced by ERß signaling in female mice at an early stage of ovarian failure.

Alterations in axon terminal NMDA receptor localization might be expected to impact various aspects of presynaptic signaling. These include calcium influx via modulating voltage-dependent calcium channels [60], nitric oxide production [31], or promoting neurotrophin signaling [61]. Significantly, NMDA receptor activation has been reliably demonstrated to influence the presynaptic release of small-molecule neurotransmitters [62]. In particular, activation of the NMDA receptor has been reported to stimulate the release of glutamate as well as GABA [62], consistent with NMDA receptor expression in excitatory and inhibitory synapses. Importantly, NMDA receptor-dependent release of these transmitters in the PVN would be predicted to have relevance for autonomic processes. Indeed, it has been shown that elevated glutamate in the PVN is associated with sympathoexcitation and hypertension [63,64], whereas PVN GABA signaling is correlated with reduced blood pressure and sympathoinhibition [65,66,67]. Further, the functional outcome of altered presynaptic NMDA receptors would be expected to be weighted by the level of receptors in excitatory versus inhibitory synapses. In this context, it is noteworthy that the present study showed that on-plasma membrane GluN1 was elevated exclusively in non-hypertensive DPN and AngII-treated peri-AOF mice. Further, axon terminals forming symmetric inhibitory-type specializations were the predominant synapse type labeled for GluN1. Therefore, these results suggest that NMDA receptors are preferentially positioned for ligand-mediated stimulation of presynaptic GABA release in DPN/AngII-treated peri-AOF mice and may provide a basis for hypertension resistance in these animals.

We have thus far considered postsynaptic and presynaptic locations separately, but postsynaptic events can significantly affect the presynapse, particularly via retrograde processes providing important feedback signaling [55]. In this context it is noteworthy that major retrograde acting molecules, including cannabinoids [68] and nitric oxide [20,69], play important roles in the PVN during hypertension via actions involving glutamate. In light of potential retrograde influences, it is noteworthy that we found in the DPN experiment in AOF mice that altered GluN1 localization was restricted to axon terminals that formed contacts. Moreover, in many instances contacts were identified synapses formed with dendritic profiles, indicating the possibility for bidirectional post and presynaptic signaling. Further, most of these dendrites expressed ERß, further suggesting that estrogen is positioned for modulating retrograde signaling [70,71] in addition to classical postsynaptic transmission. 

## 5. Conclusions and Summary

The present results along with prior reports on dendritic localization indicate distinct effects of hypertension and sex/hormonal status on NMDA receptor distribution. Plasma membrane NMDA receptors are elevated postsynaptically in PVN neurons in both hypertensive males and AOF females. However, males have higher presynaptic NMDA receptors but independently of hypertension. Interestingly, when administered an agonist of ERß, males are unresponsive, but females show increased plasma membrane NMDA receptors in axon terminals forming contacts with ERß expressing dendrites of PVN neurons. These findings are consistent with select estrogen modulation of postsynaptic to presynaptic feedback signaling at NMDA receptors, possibly involving the critical retrograde modulator nitric oxide [70,71], only in AOF females.

In sum, the present study demonstrates sex and estrogen-related divergences in the subcellular localization of the NMDA receptor in axon terminals in the PVN in response to AngII with and without hormone replacement. These findings further suggest a novel pathway by which sex and estrogen signaling may affect neural transmission in the hypothalamus by means of the presynaptic actions of NMDA receptors.

## Figures and Tables

**Figure 1 biology-13-00819-f001:**
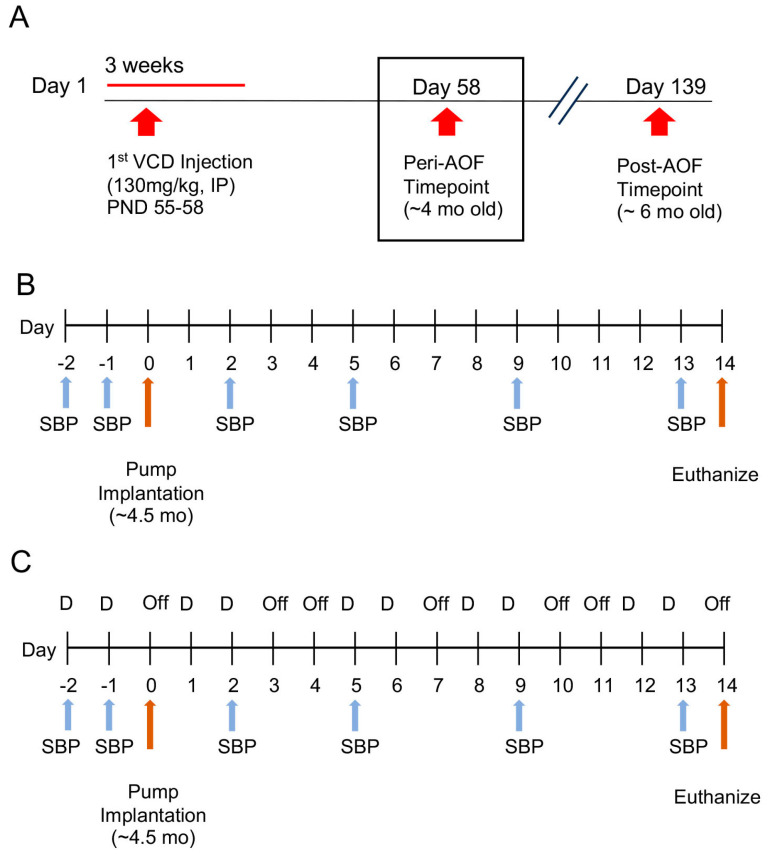
Schematic timelines of AOF induction in female mice and slow-pressor AngII infusion alone or coinciding with cyclic ERß agonist replacement. (**A**) VCD (130 mg/kg, i.p.) was injected for 3 weeks, 5 days per week starting between PND 55–58. Mice are considered peri-AOF starting 58 days following VCD injection when they are ~4 months old. (**B**) Experiment 1: Peri-AOF female mice and age-matched males were implanted with osmotic minipumps containing AngII (600 ng/kg^−1^ /min^−1^) or Sal (saline + 0.1% BSA) for 14 days. SBP was measured via tail-cuff plethysmography for 2 days preceding and 2, 5, 9 and 13 days after minipump implantation, before euthanasia on day 14 (SBP data from [8,17]). (**C**) Experiment 2: Peri-AOF female and age-matched males were implanted with osmotic minipumps containing AngII (600 ng/kg^-1^ /min^-1^) or Sal (0.1% BSA in saline) for 14 days. SBP was measured via tail-cuff plethysmography for 2 days preceding and 2, 5, 9, and 13 days after minipump implantation. The ERß agonist DPN (1 mg/kg) was administered 2 days prior to minipump implantation and continued every 2–3 days until euthanasia. (SBP data from [18]).

**Figure 2 biology-13-00819-f002:**
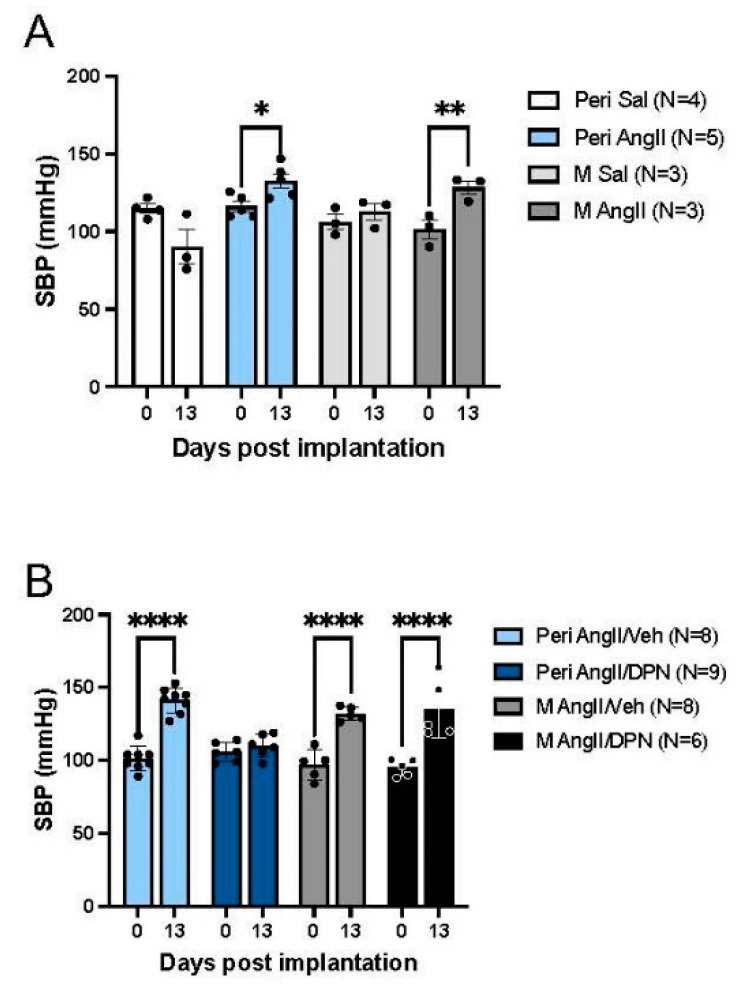
SBP in peri-AOF females and age-matched male mice infused with slow-pressor AngII and co-administered ERß agonist DPN. (**A**) Experiment 1: Peri-AOF female and male mice given AngII exhibit an increase in SBP (light blue and dark gray bars, respectively), while mice given Sal do not show increased SBP (white, gray bars). (**B**) Experiment 2: Following co-administration of cyclic DPN, peri-AOF female mice infused with AngII no longer show increased SBP (dark blue bars). However, male mice infused AngII remained hypertensive following the cyclic injection of DPN (black bars). The 0 time point corresponds to the readings taken the day before pump implantation. * *p* < 0.05; ** *p* < 0.01; **** *p* < 0.001.

**Figure 3 biology-13-00819-f003:**
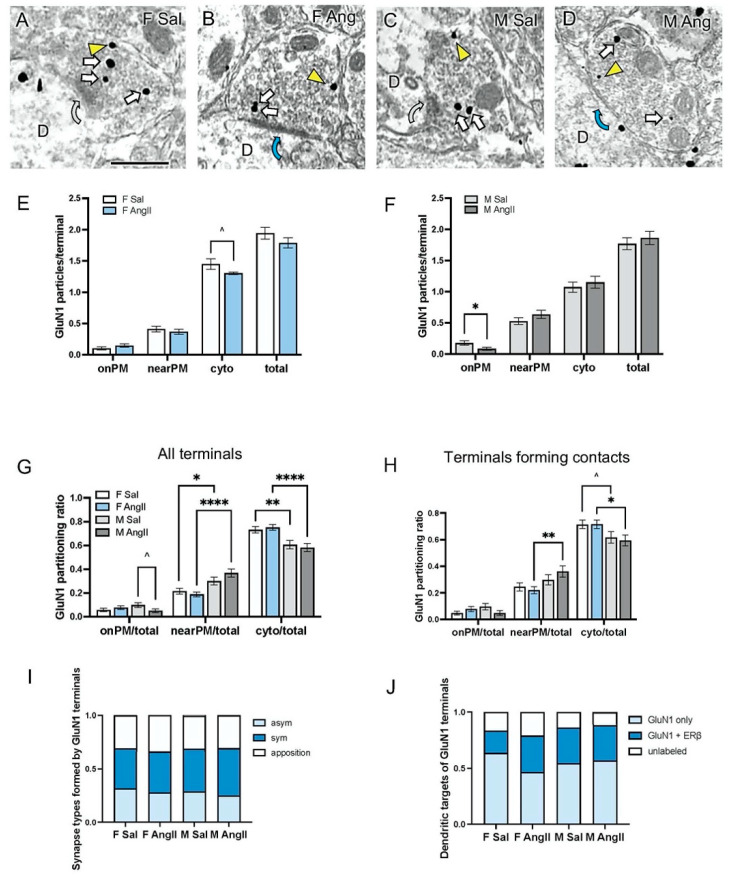
**GluN1-SIG containing terminals in males and peri-AOF females from Sal and AngII-infused mice.** (**A**–**D**) Representative electron micrographs showing the subcellular localization of GluN1-SIG particles in axon terminals of peri-AOF female and male mice infused with Sal or AngII. D = dendrite; Yellow arrowhead = nearPM SIG; arrows = cytoplasmic SIG. Curved grey arrows = appositions (**A**,**C**); Curved blue arrows = symmetric synapse (**B,D**). (**E**) Sal and AngII-infused peri-AOF female mice have similar numbers of presynaptic GluN1-SIG particles onPM, near PM and in total. AngII-infused females compared to Sal-infused females tended to have lower numbers of presynaptic GluN1-SIGs in the cytoplasm. (**F**) AngII-infused male mice showed a significant decrease in the number of presynaptic onPM GluN1-SIG particles compared to Sal-infused males. (**G**) Peri-AOF female mice had a significantly lower partitioning ratio of presynaptic nearPM GluN1-SIG compared to males. Conversely, peri-AOF females displayed a significantly greater ratio of cytoplasmic GluN1-SIG compared to males. AngII-infused peri-AOF females compared to AngII-infused males had a significantly smaller ratio of nearPM GluN1 SIGs and a significantly greater ratio of cytoplasmic GluN1-SIGs. (**H**) Of the GluN1-labeled axon terminals that formed dendritic contacts (appositions, asymmetric and symmetric synapses), AngII-infused peri-AOF females showed a significantly decreased ratio of presynaptic nearPM GluN1-SIG and a higher ratio of presynaptic cytoplasmic GluN1-SIG compared to AngII-infused males. Sal-infused peri-AOF females tended to have an increased ratio of cytoplasmic GluN1 to Sal-infused males. (**I**) Bar graph showing the proportion of terminals in each group forming appositions, asymmetric or symmetric synapse with dendrites. asym: asymmetric synapse, sym: symmetric synapse. (**J**) Bar graph showing the proportion of unlabeled, single, and dual-labeled dendritic profiles that were targets of GluN1-containing terminals. * *p* < 0.05; ** *p* < 0.01; **** *p* < 0.001, ^ *p* = 0.06.

**Figure 4 biology-13-00819-f004:**
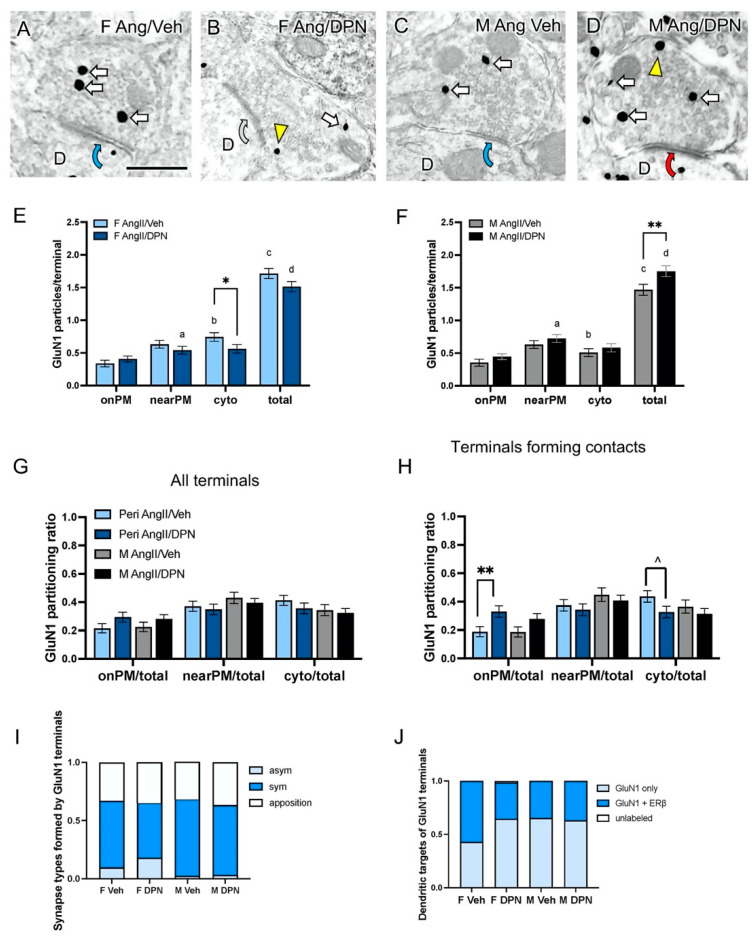
**GluN1-SIG containing terminals in males and peri-AOF females from AngII-infused mice co-administered DPN.** (**A**–**D**) Representative electron micrographs showing the subcellular localization of GluN1-SIG particles in axon terminals of AngII-infused peri-AOF female and male mice co-administered with vehicle or DPN. D = dendrite; Yellow arrowhead = nearPM SIG; arrows = cytoplasmic SIG. Curved blue arrows = symmetric synapse (**A,C**); Curved grey arrow = apposition (**B**), and Curved red arrow = asymmetric synapse (**D**). (**E**,**F**) After DPN administration, AngII-infused peri-AOF female mice showed a decreased number of cytoplasmic GluN1-SIG per terminal, while AngII-infused males had an increased total per terminal. AngII-infused peri-AOF females demonstrated a greater total number of GluN1-SIG particles per terminal as well as a greater number of cytoplasmic GluN1 SIG particles compared to males. When comparing DPN-injected peri-AOF females and matched males, there were lower numbers of near plasmalemmal and total GluN1-SIG particles. (**G**) The partitioning ratios of GluN1-SIGs showed no significant between-group differences in any cellular compartment in terminals. (**H**) Of the GluN1-labeled axon terminals that formed dendritic contacts (appositions, asymmetric and symmetric synapses), AngII-infused peri-AOF females given DPN had a significantly increased ratio of onPM GluN1 SIG particles compared to Veh females. (**I**) Bar graph showing the proportion of terminals in each group forming appostions, as well as asymmetric or symmetric synapse with dendrites. (**J**) Bar graph showing the proportion of unlabeled, single, and dual-labeled dendritic profiles targeted by GluN1-containing terminals. * *p* < 0.05; ** *p* < 0.01; ^ *p* = 0.06; a,b,c,d: *p* < 0.05.

**Figure 5 biology-13-00819-f005:**
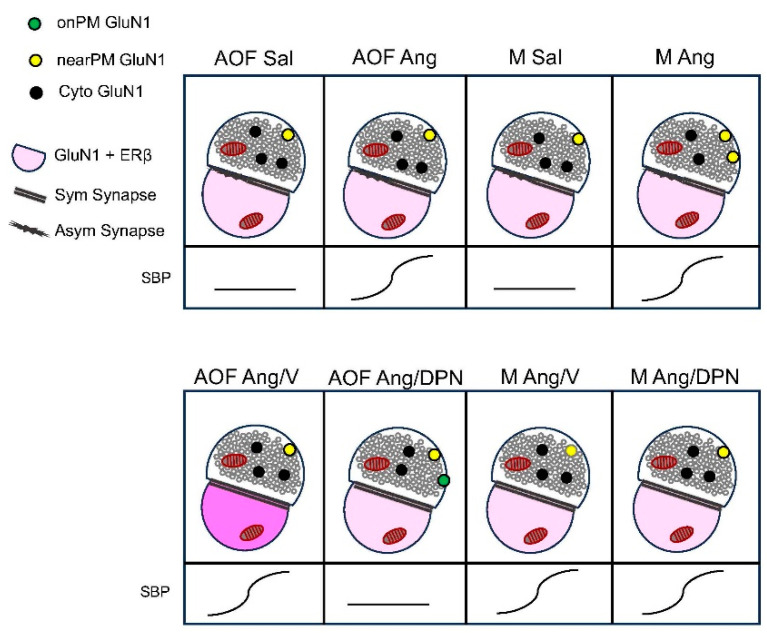
**Summary schematic.** Diagrams depicting the altered ratio of GluN1 in terminals contacting dendrites in peri-AOF females compared to males following AngII-infusion alone or co-administered cyclic DPN. **Top**: Following AngII-induced hypertension in males, GluN1 is redistributed from the cytoplasm to the near plasma membrane compartment in a mixed population of excitatory (asymmetric) and inhibitory (symmetric) synapses. **Bottom**: DPN administration to peri-AOF mice reduces AngII-induced hypertension and redistributes GluN1 from the cytoplasm to the on-plasma membrane compartment in a population of axon terminals predominantly forming inhibitory (symmetric) synapses. Vehicle-treated peri-AOF females have a greater proportion of dendrites containing both ERß and GluN1 (dark pink). SBP = systolic blood pressure.

## Data Availability

Data are available upon request. Please email Dr. Teresa Milner (tmilner@med.cornell.edu).

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
