# Peer review of "Estrogen Receptor Beta Agonist Influences Presynaptic NMDA Receptor Distribution in the Paraventricular Hypothalamic Nucleus Following Hypertension in a Mouse Model of Perimenopause"

_biology, 2024, doi:10.3390/biology13100819_

Round 1
Reviewer 1 Report
Comments and Suggestions for Authors
This manuscript utilized theaccelerated ovarian failure (AOF) animal model and immunoelectron microscopy to examine the presynaptic NMDA receptor localization during peri-AOF in female mice. Different from males males, the authors found that activation of ERß blocked hypertension and increased NMDA receptors on the membrane of presynaptic axon terminals, where it would be more available for binding of glutamate, indicating a sex-dependent action of presynaptic NMDA recruitment under the regulation of ER signaling.
The manuscript has solid scientific background, experimental design and data analysis. Only a few points need to be addressed:
1. Line 114, please make sure the font of "(www.gensat.org)" is consistent with the rest of the content.
2. In Figure 2, there's SBP measurement of mice of different treatment group at day 0 and day 13. However, according to the experimental design in Figure 1, SBP was not measure at day 0 (pump implantation). Could the authors clarify the origin of SBP measurement of Day 0?
3. In Figure 3 and 4, the Arrowhead and Arrows mark look quite similar to each other. Could it be possible to change the marking so it'd be easier for the reader to distinguish them?
Author Response
REVIEWER 1
Comment 1: Line 114, please make sure the font of "(www.gensat.org)" is consistent with the rest of the content.
Response 1: The font for www.gensat.org has been changed to be consistent with the rest of the text.
Comment 2. “In Figure 2, there's SBP measurement of mice of different treatment group at day 0 and day 13. However, according to the experimental design in Figure 1, SBP was not measure at day 0 (pump implantation). Could the authors clarify the origin of SBP measurement of Day 0?”
Response 2: The 0 timepoint corresponds to the readings taken the day before pump implantation. This has been clarified in the figure 2 legend.
Comment 3. In Figure 3 and 4, the Arrowhead and Arrows mark look quite similar to each other. Could it be possible to change the marking so it'd be easier for the reader to distinguish them?
Response 3. The arrowhead was filled in with yellow so that it could be more easily distinguished from the white arrows. New versions of figures 3 and 4 have been uploaded.
Editor's comments:
Comment 1: Please change the reference style as "[1]" like.
Response 1: The reference style has been changed as requested.
Comment 2: Please provide the ethic code.
Response 2: The following statement has been added: The submitted manuscript conforms to MDPI’s publication ethics. We fully adhere to its Core Practices and to its guidelines.
Comment 3: Please complete the back matters including "Institutional Review Board Statement", "Data Availability Statement", "Informed Consent Statement". Otherwise, at the Acknowledgements you mentioned "funding", should it be in "Funding" part? Please check.
Response 3: The IRB, Data Availability and Informed Consent statements have been added. “Acknowledgements” has been changed to “Funding”.
Comment 4: For the Figure 1 you used, we noticed that you have references, please provide the copyright.
Response 4: The systolic blood pressure (SBP) data from 3 references were used in this manuscript. Dr. Milner is a primary author on all 3 manuscripts. Copyrights were obtained from Journal of Comparative Neurology and Neuroendocrinology. A copyright was not required from the Journal of Neuroscience. Proof of copyright has been provided.
Reviewer 2 Report
Comments and Suggestions for Authors
This manuscript sheds light on the role of glutamatergic receptor distribution in females susceptible to hypertension during menopause and explores how estrogen receptors may normalize these alterations. The study is well-designed and provides significant insights, contributing valuable knowledge to enhance our understanding of female susceptibility to neurological impairments.
- The authors examined the localization of the PM and the cytoplasmic localization of the GluN1 subunit receptor. However, the localization of other NMDA receptor subunits was not investigated. Could the authors provide insights into why these additional subunits were not examined?
- The study might more informative from measuring changes in spine length or dendritic length. Including these experiments could increase reviewers' interest and improve the overall comprehensiveness of the study.
- The discussion is well-written. However, incorporating more mechanistic insights into how GluN1 influences plasticity or synaptic depression/alteration in the context of hypertension could further enhance the depth and comprehensiveness of the results.
- Males have lower estrogen levels compared to females. It would be useful to discuss why the coordination between estrogen and GluN1 remains unaffected despite this difference. Providing additional information on this topic in the discussion section could improve the understanding of the research.
- Overall, the manuscript is highly informative and provides valuable insights into hypertension in aged females.
Author Response
REVIEWER 2
Comment 1. The authors examined the localization of the PM and the cytoplasmic localization of the GluN1 subunit receptor. However, the localization of other NMDA receptor subunits was not investigated. Could the authors provide insights into why these additional subunits were not examined?
Response 1: GluN1 was investigated because it is essential for making most NMDA receptor variants. Further, most prior investigations of the subcellular localization of NMDA receptors have used GluN1 as a stand in for this receptor, thus facilitating comparisons across studies. This has been noted in the revised manuscript (Results page 8, lines 304-307).
Comment 2. The study might be more informative from measuring changes in spine length or dendritic length. Including these experiments could increase reviewers' interest and improve the overall comprehensiveness of the study.
Response 2. The subcellular localization of GluN1 in the present study was assessed in axon terminals, rather than dendrites or dendritic spines. Unlike dendrites, axon terminals do not considerably vary in morphological properties and therefore they were not measured.
Comment 3. The discussion is well-written. However, incorporating more mechanistic insights into how GluN1 influences plasticity or synaptic depression/alteration in the context of hypertension could further enhance the depth and comprehensiveness of the results.
Response 3. We have further elaborated the discussion of pre and postsynaptic mechanisms of NMDA receptor signaling and hypertension in the revised manuscript (Discussion and Summary, page 15, lines 580-589)
Comment 4. Males have lower estrogen levels compared to females. It would be useful to discuss why the coordination between estrogen and GluN1 remains unaffected despite this difference. Providing additional information on this topic in the discussion section could improve the understanding of the research.
Response 4. We have added details on the relationship between estrogen and NMDA signaling in the revised manuscript (Discussion page 14 lines 538-547).